# Approaching a fully-polarized state of nuclear spins in a solid

Peter Millington-Hotze [1], Harry E. Dyte[1], Santanu Manna[2,3],
Saimon F. Covre da Silva [2], Armando Rastelli [2] & Evgeny A. Chekhovich [1] ✉

Magnetic noise of atomic nuclear spins is a major source of decoherence in solid-state spin qubits. In theory, near-unity nuclear spin polarization can eliminate decoherence of the electron spin qubit, while turning the nuclei into a useful quantum information resource. However, achieving sufficiently high nuclear polarizations has remained an evasive goal. Here we implement a nuclear spin polarization protocol which combines strong optical pumping and fast electron tunneling. Nuclear polarizations well above 95% are generated in GaAs semiconductor quantum dots on a timescale of 1 minute. The technique is compatible with standard quantum dot device designs, where highly-polarized nuclear spins can simplify implementations of qubits and quantum memories, as well as offer a testbed for studies of many-body quantum dynamics and magnetism.

The capability of initializing a quantum system into a well-defined eigenstate is one of the fundamental requirements in quantum science and technology. This has been demonstrated for individual and dilute nuclear spins in the solid state[1,2], but remains a long-standing challenge for dense three-dimensional lattices of nuclear spins. For the quantum ground state of a spin ensemble the polarization degree is $P_N = \pm 100\%$, which is equivalent to absolute zero spin temperature. Very high polarizations, $P_N \approx 95{-}99\%$, can be reached through brute-force cooling in certain bulk materials, but the cooling cycle may take hours or even days[3,4]. More scalable approaches seek to use individual or dilute electron spins to polarize the dense nuclear ensembles. Microwave pumping of paramagnetic impurities in bulk solids[5,6] provides polarizations up to $P_N \approx 80{-}90\%$. In semiconductor nanostructures, $P_N \approx 50{-}80\%$ is achieved either through electronic transport[7] or optical excitation[8]. However, polarizations much closer to unity are needed to suppress the electron spin qubit dephasing, whose rate scales as $\sqrt{1 - P_N^2}$[9], or reduce the nuclear ensemble entropy, which scales as $\frac{1-P_N}{2}(1 - \ln(\frac{1-P_N}{2}))$[10]. Therefore, different techniques are needed to approach a fully-polarized nuclear state.

Extensive theoretical studies have been conducted to understand what limits nuclear spin pumping in a central-spin scenario, where the electron can be polarized on demand, while the ensemble of $N$ nuclei can only be accessed through hyperfine (magnetic) coupling with that central electron (Fig. 1a). The formation of coherent "dark" states[11] has been shown to suppress the transfer of polarization from the electron to nuclei[12]. Thus an open question remains – is it possible, even in principle, to reach a fully-polarized nuclear state in a real central-spin system?

We work with GaAs/AlGaAs quantum dots (QDs) and use optical excitation to polarize nuclear spins. While the optical method is well known[13], achieving near-unity polarizations and understanding the underlying physics proved challenging. Here, we show that the solution is to combine strong optical excitation with fast carrier tunneling, which resolves the main bottleneck of slow optical recombination. Moreover, no "dark"-state limitation occurs, which we also attribute to the extremely short lifetime of the electron spin. As a result, we achieve nuclear polarization degrees well above $P_N > 95\%$. The maximum polarizations vary between individual QDs, which we ascribe to slight fluctuations in QD shapes and partial relaxation of the optical selection rules. For the best dots we derive $P_N \gtrsim 99\%$, limited only by the accuracy of the existing measurement techniques. These high polarizations surpass the predicted $P_N \gtrsim 90\%$ threshold for achieving extended electron spin qubit coherence[14,15], quantum memory operation[15,16],

[1]Department of Physics and Astronomy, University of Sheffield, Sheffield S3 7RH, United Kingdom. [2]Institute of Semiconductor and Solid State Physics, Johannes Kepler University Linz, Altenberger Str. 69, Linz 4040, Austria. [3]Present address: Department of Electrical Engineering, Indian Institute of Technology Delhi, New Delhi 110016, India. ✉e-mail: e.chekhovich@sheffield.ac.uk

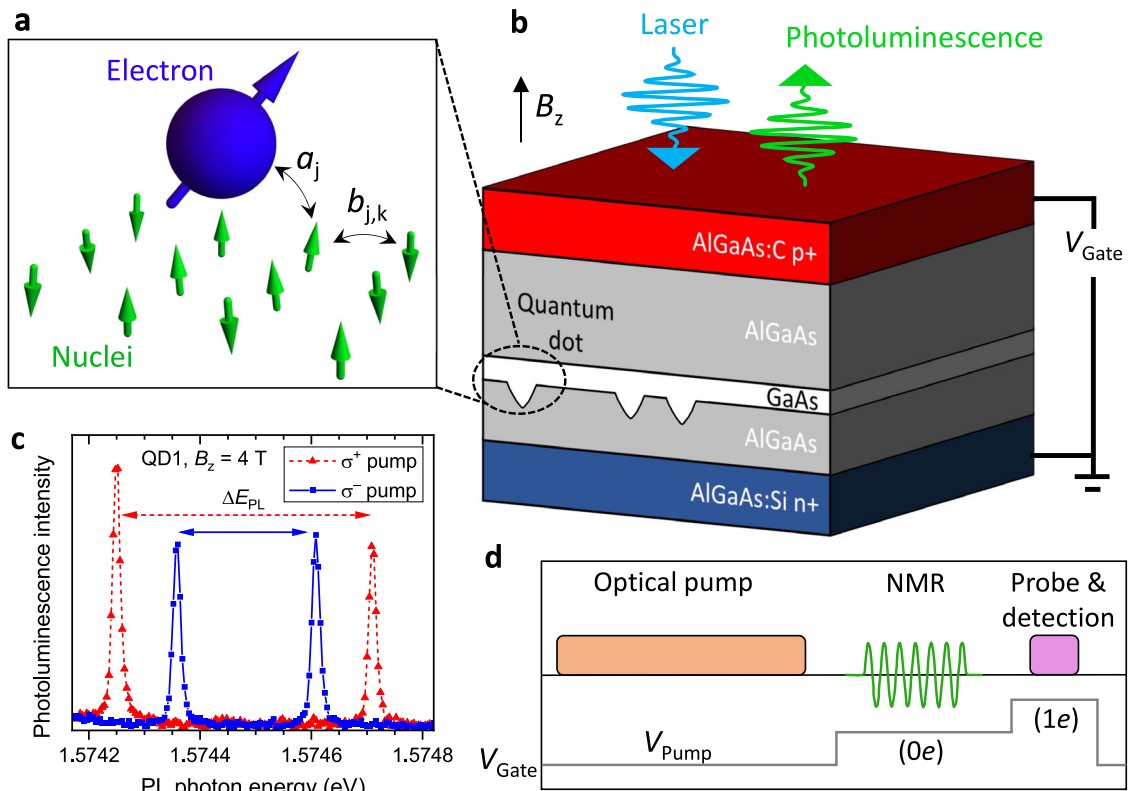

**Fig. 1 | Optical control of quantum dot nuclear spins. a** Schematic of a central electron spin and an ensemble of nuclear spins coupled through hyperfine interaction with constants $a_j$. The nuclei are coupled through dipolar interactions with pairwise constants $b_{j,k}$ (see Supplementary Note 2). **b** Schematic cross-section of a $p-i-n$ diode with embedded epitaxial GaAs quantum dots. Laser excitation, photoluminescence collection and the external magnetic field are directed along the sample growth axis $z$. Doped semiconductor layers are used to apply the gate bias $V_{Gate}$, resulting in a tunable electric field along $z$. **c** Typical photoluminescence spectra of a negatively charged trion $X^-$ in an individual QD. The spectral splitting $\Delta E_{PL}$ depends both on magnetic field $B_z$ and the helicity ($\sigma^+$ or $\sigma^-$) of the optical pumping due to the buildup of the nuclear spin polarization. **d** Experimental cycle consisting of nuclear spin optical pumping, nuclear magnetic resonance (NMR) excitation, and optical probing of the photoluminescence spectrum. $V_{Gate}$ is switched between an arbitrary level $V_{Pump}$ and the levels that tune the QD into the electron-charged ($1e$) and neutral ($0e$) states.

superradiant electron-nuclear spin dynamics[17,18], as well as magnetic-ordering phase transition[19,20].

## Results

The semiconductor device, sketched in Fig. 1b, is a $p-i-n$ diode with epitaxial GaAs QDs embedded into the AlGaAs barrier layers (see Supplementary Note 1). By changing the gate bias $V_{Gate}$ it is possible to charge the QD with individual resident electrons[21,22] and apply a tunable electric field. Each individual QD contains $N \approx 10^5$ nuclei, with the three abundant isotopes $^{75}$As, $^{69}$Ga and $^{71}$Ga, all possessing spin momentum $I = 3/2$. The sample is cooled to $\approx 4.25$ K and placed in a magnetic field $B_z$ parallel to the electric field and sample growth direction (see Supplementary Note 3). Thanks to the selection rules[13], optical excitation creates spin-polarized electron-hole pairs; $\sigma^+$ polarized photons with $\pm 1$ angular momentum (in units of $\hbar$) generate electrons with spin projection $s_z = \mp 1/2$. Owing to the electron-nuclear hyperfine interaction (Fig. 1a), a polarized electron can transfer its spin to one of the nuclei and, through repeated optical pumping, induce a substantial polarization $|P_N|$. Conversely, the energy of the photon emitted from electron-hole recombination depends on the mutual alignment of the electron spin and the total magnetic field, which is a sum of $B_z$ and the effective field of the polarized nuclei. The resulting optical spectrum is a doublet (Fig. 1c), whose splitting $\Delta E_{PL}$ is used as a sensitive probe of the nuclear spin polarization state. We define the exciton hyperfine shift $E_{hf} = -(\Delta E_{PL} - \Delta E_{PL,0})$, where $\Delta E_{PL,0}$ is the splitting measured for depolarized nuclei ($P_N \approx 0$).

The high resolution optical spectra (Fig. 1c), required to measure $E_{hf}$, can only be observed for a narrow range of gate biases and optical excitation powers. Therefore, we use a pump-probe technique (Fig. 1d), where the nuclear spins are polarized by an optical pump with an arbitrary set of parameters, while the optical probe parameters are fixed and optimized for the readout of $E_{hf}$. Conducting experiments at high magnetic field $B_z = 10$ T, we maximize the hyperfine shift $|E_{hf}|$ by optimizing the following parameters: the elliptical polarization of the optical pump, its power $P_{Pump}$, photon energy $E_{Pump}$ and the bias $V_{Pump}$ during pumping. The results are interpreted with reference to photoluminescence data. Figure 2a shows low power luminescence spectra, which reveal a well-known bias-controlled charging of the ground state ($s$-shell) exciton in a QD[22]. High optical power (Fig. 2b) broadens the spectra, also populating the higher shells $p$, $d$, etc.[23,24]. (See additional data in Supplementary Note 4).

The dependence of nuclear-induced shift $E_{hf}$ on $E_{Pump}$ and $V_{Pump}$, shown in Fig. 2c, reveals spectral bands that match the excitonic shells in Fig. 2b, demonstrating that nuclear spin pumping proceeds through resonant optical driving of the QD exciton transitions. The largest $|E_{hf}|$ is observed when the pump is resonant with the $s$ shell ($E_{Pump} \approx 1.565$ eV), and at a large reverse bias $V_{Pump} = -2.3$ V, where photoluminescence is completely quenched. Moreover, the optimal pump laser power $P_{Pump} = 1.5$ mW is five orders of magnitude higher than the saturation power of the $s$-shell luminescence. Based on these observations, the nuclear spin pumping effect can be understood as a cyclic process sketched in Fig. 2d. First, circularly-polarized resonant

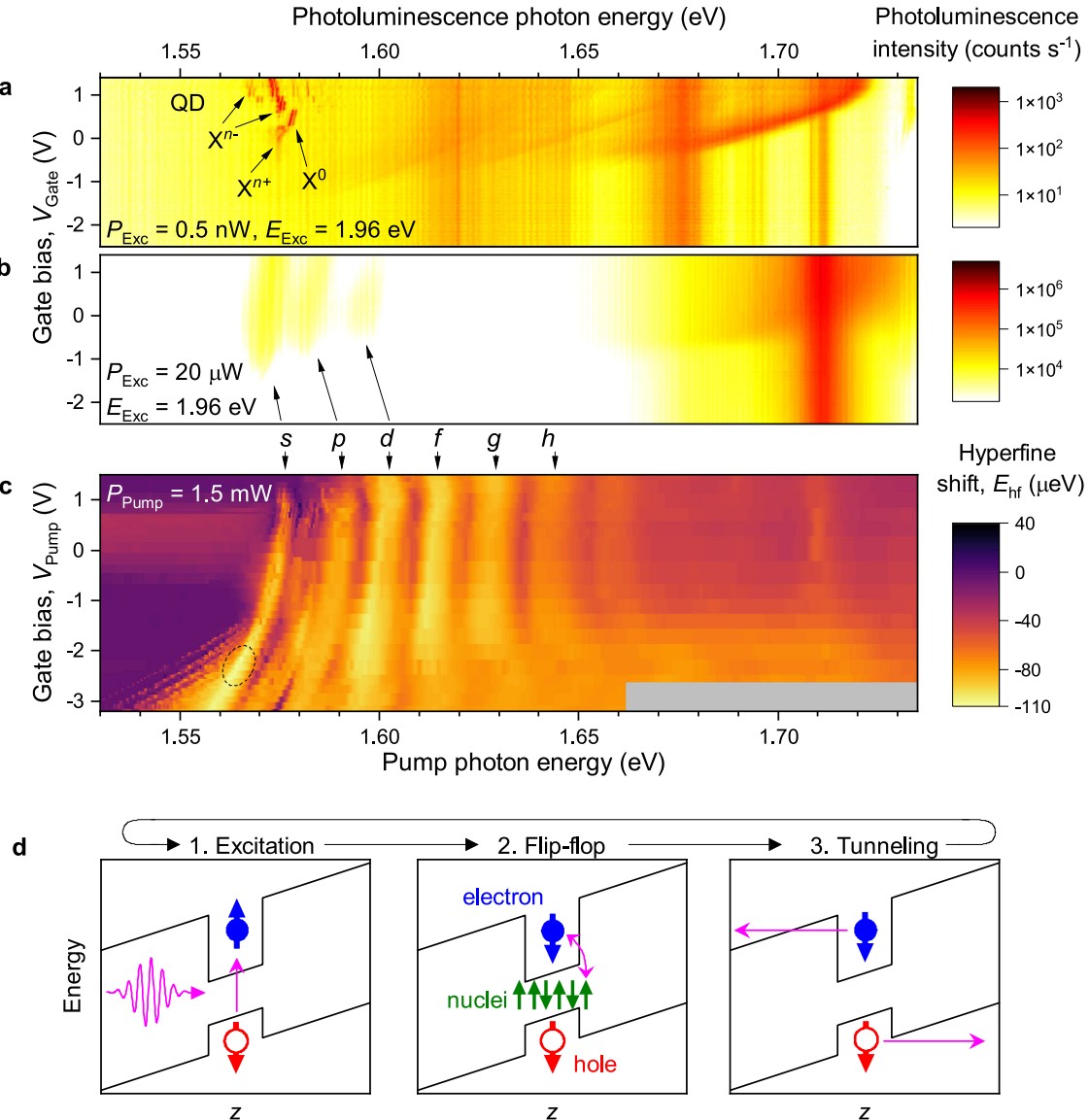

**Fig. 2 | Tunneling-assisted optical nuclear spin pumping. a** Bias-dependent photoluminescence spectra of an individual dot QD1 measured at $B_z = 10$ T at low excitation power $P_{Exc} = 0.5$ nW and excitation photon energy $E_{Exc} = 1.96$ eV. Labels show neutral ($X^0$), positively ($X^{n+}$) and negatively ($X^{n-}$) multi-charged QD excitons. Broad spectral features at higher energies arise from the AlGaAs layers. **b** Photoluminescence spectra at an increased power $P_{Exc} = 20\,\mu$W reveal saturated emission from higher QD exciton shells, labeled $s$, $p$, $d$. **c** Hyperfine shift measured in a pump-probe experiment (Fig. 1d) on QD1 as a function of gate bias $V_{Pump}$ and

the photon energy $E_{Pump}$ of the $\sigma^+$ polarized optical pump with power $P_{Pump} = 1.5$ mW. Parameter regions where no data has been measured are shown in gray. Excitonic spectral features are labeled up to the $h$ shell. The dashed ellipse highlights the parameters that result in the most efficient nuclear spin polarization. **d** Schematic of the conduction and valence band edges along the $z$ direction, as well as confined electron (full circles) and hole (open circles) states. The three stages of the cyclic nuclear spin pumping process are shown schematically. Source data for **a**–**c** are provided as a Source Data file.

optical excitation creates a spin-polarized electron-hole pair in the quantum dot. Then, the electron has a small but finite probability to undergo a flip-flop with one of the nuclei, increasing the ensemble polarization $|P_N|$. Finally, in order to proceed to the next cycle, the electron is removed through tunneling. The tunneling time, estimated from bias-dependent photoluminescence in Supplementary Note 4, is $\lesssim 0.1$ ps, much shorter than the $\approx 300$ ps radiative recombination time[25]. The combination of high-power optical pumping and fast tunneling escape results in rapid cycling. This in turn leads to a high rate of nuclear spin pumping, which helps to outpace the inevitable nuclear spin relaxation. The cycling time is also much shorter than the period of coherent electron precession $\gtrsim 20$ ps, ensuring the spin-flipped electrons are removed before they can undergo a reverse flip-flop[26]. The ultimate result is a large steady-state nuclear hyperfine shift $|E_{hf}| > 110\,\mu$eV, exceeding $|E_{hf}|$ observed previously[8,27].

Although $E_{hf}$ scales linearly with nuclear polarization degree $P_N$, its absolute value depends on the QD structure. The electron wavefunction leaks into the barriers where the fraction of Ga atoms replaced with Al atoms is not known precisely. A more reliable measurement of the $P_N$ is achieved through nuclear magnetic resonance (NMR) spin thermometry (see Supplementary Note 5 for details). The method assumes Boltzmann probability distribution $p_m \propto e^{m\beta}$ for each nucleus to occupy a state with spin projections $m$, where $\beta$ is the dimensionless inverse spin temperature. For spin $I = 1/2$, where $m = \pm 1/2$, any statistical distribution has the Boltzmann form. By contrast, for $I > 1/2$, the Boltzmann distribution expresses the non-trivial nuclear spin temperature hypothesis[28], verified for epitaxial GaAs quantum dots previously[8].

In order to perform spin thermometry, we first measure the single-QD NMR spectra[29], as exemplified in Fig. 3a for $^{69}$Ga spins. The

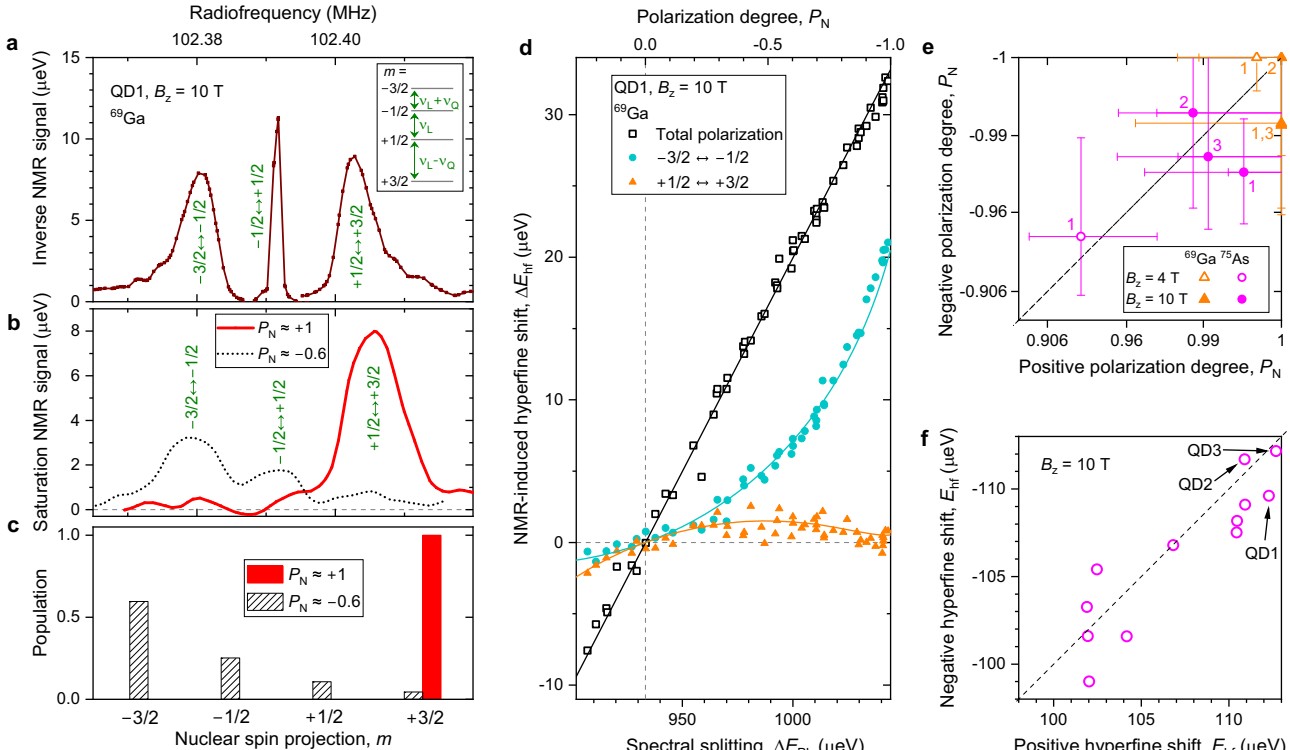

**Fig. 3 | Nuclear magnetic resonance spin thermometry. a** High-resolution spectrum of $^{69}$Ga measured in QD1 at $B_z = 10$ T using "inverse NMR" signal enhancement technique[29]. Inset shows the nuclear energy levels with spin projections $m = \pm 1/2, \pm 3/2$. The resonance between $m = \pm 1/2$ is at a pure Larmor frequency $\nu_L$, whereas the satellite transitions involving $m = \pm 3/2$ are split off by the quadrupolar shifts $\mp\nu_Q$ (for Ga nuclei the shifts are predominantly negative $\nu_Q < 0$). **b** Low-resolution spectrum of the same QD1, but measured using the saturation technique in order to reveal the population probabilities of the nuclear spin levels. **c** Population probabilities of spin levels with different $m$, sketched for the same two nuclear polarization degrees as in **b**. **d** Hyperfine shift variation arising from selective NMR manipulation of the $^{69}$Ga nuclear spins plotted against the initial photoluminescence spectral splitting $\Delta E_{PL}$, varied by changing the optical pump

wavelength and polarization. Squares show the total $^{69}$Ga hyperfine shift measured by broadband saturation of the entire NMR triplet, which equalizes populations $p_m$ for all $m$. Circles and triangles show the selective signals of the $\pm 1/2 \leftrightarrow \pm 3/2$ resonances measured via frequency-swept adiabatic inversion. Lines show fitting, from which nuclear spin polarization degree is derived and plotted in the top horizontal scale (see Supplementary Note 5). **e** Maximum positive and minimum negative nuclear spin polarization degrees $P_N$ derived for $^{69}$Ga (triangles) and $^{75}$As (circles) in individual dots QD1 - QD3 (indexed by numbers adjacent to the symbols) at $B_z = 10$ T (solid symbols) and $B_z = 4$ T (open symbols). Error bars are 95% confidence intervals. **f** Maximum positive and minimum negative hyperfine shifts measured on individual dots QD1–QD12 at $B_z = 10$ T. Source data for **d** and **f** are provided as a Source Data file. Source data for **e** can be found in Supplementary Information.

three magnetic-dipole transitions of the 3/2 spins are well resolved thanks to the quadrupolar shifts $\nu_Q$, which originate from the lattice mismatch of GaAs and AlGaAs. Compared to the Larmor frequency $\nu_L \approx 100$ MHz, these strain-induced quadrupolar effects $|\nu_Q| \lesssim 100$ kHz are still too small to impede nuclear spin pumping. This is a significant advantage over the highly-strained Stranski–Krastanov QDs[29], where $|\nu_Q| \approx 1$–10 MHz so that large $|P_N|$ is prohibited simply because nuclear eigenstates are not aligned along the magnetic field[30] (the misalignment is characterized by the ratio $\propto (\nu_Q/\nu_L)^2$). The resolved NMR triplet is essential, as it allows $\beta$ to be derived from the Boltzmann exponent, which then relates to $P_N$ through the standard Brillouin function. Qualitatively this is demonstrated in Fig. 3b with simple saturation NMR spectroscopy[31]. At moderate polarization $P_N \approx -0.6$ (dashed line) all three magnetic-dipole transitions $m \leftrightarrow m+1$ are observed, and their amplitudes are proportional to the differences $|p_{m+1} - p_m|$ (Fig. 3c). At the maximum positive polarization (solid line) a single NMR peak $+1/2 \leftrightarrow +3/2$ is observed, indicating that nearly all spins have been cooled to the $m = +3/2$ state.

For quantitative spin thermometry we measure the peak areas of the $-3/2 \leftrightarrow -1/2$ and $+1/2 \leftrightarrow +3/2$ NMR transitions at different nuclear polarizations. The results are shown in Fig. 3d (circles and triangles), together with the total signal obtained by saturating all three NMR transitions (squares). We take into account the small overlaps of the NMR triplet components (see Supplementary Note 5) and use Boltzmann model fitting (lines) to derive the polarization

degree $P_N$ (top axis). The model reproduces well both the linear dependence of the total NMR signal and the non-linear dependencies of the selective $\pm 1/2 \leftrightarrow \pm 3/2$ signals, revealing a close approach to $P_N \approx -1$. Qualitatively, at $P_N = -1$ the $m = +1/2, +3/2$ states must be depopulated, resulting in a vanishing $+1/2 \leftrightarrow +3/2$ signal, as indeed observed experimentally. Moreover, at $P_N = -1$ the $-3/2 \leftrightarrow -1/2$ signal must be 2/3 of the total NMR signal, also in good agreement with experiment. By switching from $\sigma^+$ to $\sigma^-$ optical pumping we also approach $P_N \approx +1$. The largest positive and negative $P_N$ are shown in Fig. 3e for individual dots QD1–QD3, chosen for their highest $|E_{hf}|$. At the highest static field $B_z = 10$ T the best-fit estimates for $^{69}$Ga are around $|P_N| \approx 0.99$, with somewhat lower $|P_N| \approx 0.98$ for $^{75}$As. With 95% confidence, $|P_N|$ exceeds 0.95, but the data is also compatible with $|P_N| = 1$. It is thus possible that the actual polarization is much closer to unity – at present, the measurement accuracy is the main limitation. Spin thermometry on one of the QDs at $B_z = 4$ T yields similarly high polarizations $|P_N| \gtrsim 0.93$, although the measurement accuracy is reduced due to the less efficient optical probing.

A simpler measurement of the largest positive and negative hyperfine shift $E_{hf}$ is shown in Fig. 3f for 12 randomly chosen dots. For some QDs, nuclear polarization does not exceed $|P_N| \approx 0.9$. We also observe that for all studied QDs the optimal optical polarization of the pump is not circular, but is rather elliptical[32], with a randomly-oriented linearly-polarized contribution ranging between 0 and 0.4 (see Supplementary Note 3). This points to heavy-light hole mixing, which is

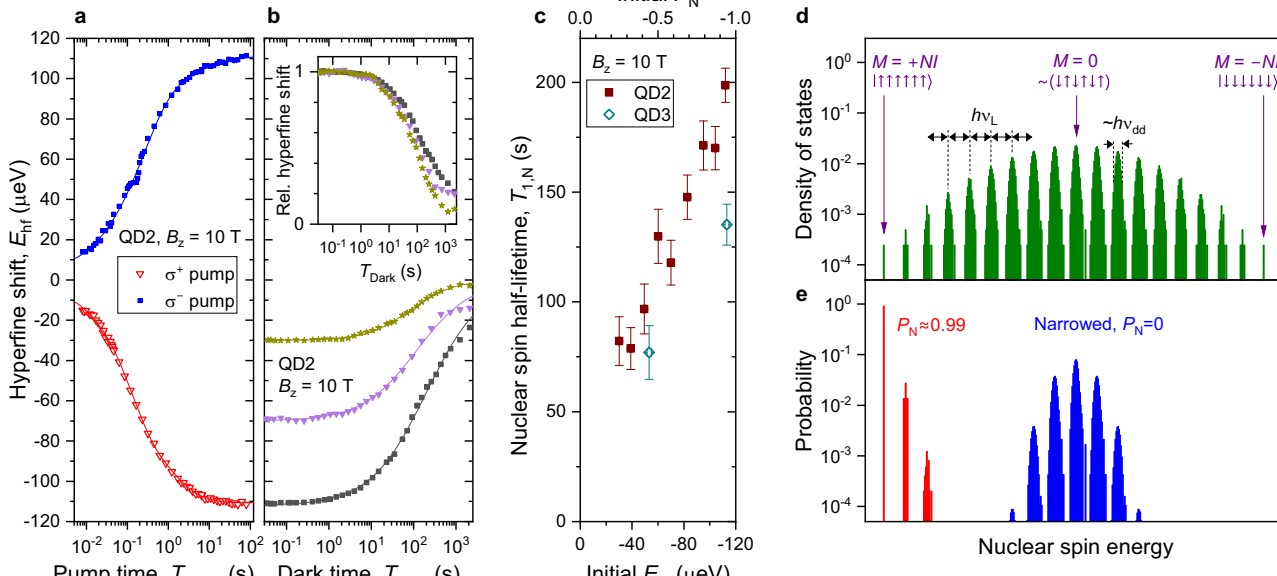

**Fig. 4 | Nuclear spin dynamics. a** Nuclear spin buildup dynamics measured (symbols) in an individual dot QD2 at $B_z = 10$ T under $\sigma^+$ (triangles) and $\sigma^-$ (squares) optical pumping. Lines show biexponential fitting. **b** Nuclear spin relaxation dynamics in the dark measured in a neutral QD state following $\sigma^+$ optical pumping (squares). The same relaxation dynamics are also measured with partial NMR saturation after the optical pump, which reduces the initial $P_N$ (triangles and stars). Lines show fitting used to derive the nuclear spin half-lifetimes $T_{1,N}$. Inset shows the same data, but normalized by the initial hyperfine shift at short dark times $T_{Dark}$. **c** Nuclear spin relaxation times $T_{1,N}$ as a function of the initial hyperfine shift $E_{hf}$. The corresponding approximate initial $P_N$ is shown on the top axis. Error bars are 95% confidence intervals. **d** Density of states calculated for $N = 6$ dipolar-coupled $I = 3/2$ nuclei (without the electron). Each band, broadened by dipolar couplings $h\nu_{dd} \propto \max |b_{j,k}|$, corresponds to a well-defined total spin projection $M$. The adjacent bands are split by the Zeeman energy $h\nu_L$. **e** Population probability of the eigenstates, calculated for the spectrum in **d** and for two types of mixed states: Boltzmann distribution of Zeeman energies with high polarization $P_N \approx 0.99$ (red) and a narrowed Gaussian distribution with $P_N = 0$ (blue). Source data for **a**–**c** are provided as a Source Data file.

always present in QDs[33] and is more pronounced under low-symmetry confinement[34,35]. In additional measurements, where the symmetry is reduced on purpose through uniaxial stress or tilting the magnetic field by $\approx 12°$, we indeed find a significant reduction in maximum $|E_{hf}|$. Therefore, the dot-to-dot variation of $P_N$ is attributed to the randomness of the QD morphology.

The buildup dynamics, measured under optimal nuclear spin pumping, are shown in Fig. 4a. The approach to the steady state is non-exponential since the nuclei that are further away from the center of the QD are less coupled to the electron and take longer to polarize. It takes on the order of $\approx 60$ s to reach the steady-state $P_N$ within the measurement accuracy. Once optical pumping is switched off, nuclear spins depolarize in the dark (squares in Fig. 4b) on a timescale of minutes, mainly through spin diffusion[36]. Such long lifetimes mean that a highly-polarized nuclear spin state can be prepared and used to extend electron spin qubit coherence over a large number of short (few nanoseconds) qubit operations. We further examine the effect of the initial $P_N$ on the relaxation dynamics by augmenting the optically-pumped nuclear state with a short partially-depolarizing NMR pulse (triangles and stars in Fig. 4b). When normalized by the initial polarization, the plot reveals accelerated nuclear spin relaxation under reduced initial polarization (inset in Fig. 4b). This is quantified in Fig. 4c, where at high polarization the nuclear spin relaxation half-lifetime $T_{1,N}$ is seen to be a factor of $\approx 2$–3 longer than in case of low initial polarization (the lowest studied initial polarization is limited by the accuracy of the $T_{1,N}$ measurement). This is a non-trivial result: scaling of the initial $P_N$ should not change $T_{1,N}$ within the linear spin diffusion model.

In order to explain the non-linear diffusion, we consider the eigenspectrum of a nuclear spin ensemble, with an example shown in Fig. 4d for $N = 6$ spins $I = 3/2$. The adjacent bands are separated by the large Zeeman energy $h\nu_L$ (typical $\nu_L \approx 100$ MHz at $B_z = 10$ T), which corresponds to a flip of a single nucleus, accompanied by a $\pm 1$ change

in the total ensemble spin projection $M$. Each band consists of all possible superpositions with a given $M$, with degeneracy lifted by the small ($\nu_{dd} \approx 1$ kHz) nuclear-nuclear dipolar magnetic interaction. For $M \approx 0$ (i.e. $P_N \approx 0$) the broadening of each band is maximal, characterized by the dipole-dipole energy $h\nu_{dd}$. With $|P_N|$ approaching unity, the distribution of the available dipolar energies narrows, eventually vanishing for the two fully-polarized states with $M = \pm NI$ (i.e. $P_N = \pm 1$). The dipolar reservoir can act as a source or sink of energy for a flip-flop spin exchange between two nuclei whose energy gaps are slightly different (for example due to the inhomogeneity of the quadrupolar shifts $\nu_Q$). Nuclear spin diffusion proceeds through such flip-flops. Therefore, the slow-down of diffusion at high initial $|P_N|$ is well explained by the narrowing of the dipolar reservoir.

The narrowing of the nuclear dipolar reservoir is conceptually similar to the state-narrowing technique, which aims to reduce the statistical dispersion of the nuclear Zeeman energies $\propto M$ in order to enhance the coherence of the electron spin qubit. An example of a narrowed mixed state is sketched in Fig. 4e for $P_N \approx 0$, but with uncertainty in $M$ reduced down to a few units, as demonstrated experimentally previously[37,38]. The fundamental advantage of a polarized state (also sketched in Fig. 4e), is that it not only narrows the uncertainty in $M$ by a factor $\propto \sqrt{1 - P_N^2}$ (see derivation in Supplementary Note 4D and Supplementary Data 1 and 2), but also reduces the dipolar broadening. In other words, our scheme represents true cooling with $|P_N| \to 1$, whereas the narrowing schemes can be seen as partial cooling of certain degrees of freedom of the nuclear ensemble. The ultimate limit of $P_N = \pm 1$ corresponds to the only two non-degenerate nuclear states, for which the electron spin qubit coherence is predicted to be essentially non-decaying[14,15]. By contrast, even if the dispersion of $M$ is reduced to zero, the dipolar energy uncertainty of a depolarized ensemble may still cause dynamics on the timescales of $1/\nu_{dd} \approx 1$ ms, leading in turn to electron spin qubit decoherence.

Investigation of electron spin coherence in a highly-polarized nuclear spin environment is an interesting subject for future work and may also provide a more sensitive tool for nuclear spin thermometry near $|P_N| \approx 1$. Alternatively, more accurate measurement of $P_N$ can be sought through "trigger" detection method[28], relying on nuclear-nuclear interactions.

## Discussion

Large nuclear polarizations are achieved here on a standard $p - i - n$ diode device, fully compatible with high-quality electron spin qubit operation, as demonstrated recently in the same semiconductor structure[39]. The technique is simple to implement and robust – once optical pumping parameters are optimized for a certain QD, they do not require any correction over months of experiments. Even larger nuclear polarizations can be sought by combining QDs of high in-plane symmetry with biaxial strain in order to reduce the heavy-light hole mixing. Our nuclear spin cooling method uses the purity of the optical pump polarization as the final heat sink, ultimately limiting the achievable $P_N$. This is different from the resonant "dragging" schemes[40–42] where the ultimate heat sink is the photon number in the optical mode, offering in principle a much closer approach to $|P_N| \approx 1$, provided the dark-state bottleneck could be avoided. Combining the advantages of the two approaches in a two-stage cooling cycle can be a route towards the ultimate goal of initializing a nuclear spin ensemble into its fully-polarized quantum ground state. This would be a pre-requisite for turning the enormously large Hilbert space of the $N \approx 10^5$ QD nuclei into a high-capacity quantum information resource.

## Data availability

The key data generated in this study are provided in the Source Data file SourceData.zip. The rest of the data that support the findings of this study are available from the corresponding author upon request. Source data are provided with this paper.

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

## Acknowledgements

P.M-H. and H.E.D. were supported by EPSRC doctoral training grants. E.A.C. was supported by a Royal Society University Research Fellowship and EPSRC award EP/V048333/1. A.R. acknowledges support of the Austrian Science Fund (FWF) via the Research Group FG5, I 4320, I 4380, I 3762, the Linz Institute of Technology (LIT), and the LIT Secure and Correct Systems Lab, supported by the State of Upper Austria, the European Union's Horizon 2020 research and innovation program under Grant Agreements No. 899814 (Qurope), No. 871130 (Ascent+), the QuantERA II project QD-E-QKD and the FFG (grant No. 891366).

## Author contributions

S.M., S.F.C.S. and A.R. developed, grew, and processed the quantum dot samples. P.M-H. and E.A.C. conducted nuclear spin pumping experiments. H.E.D. and E.A.C. conducted supporting experiments on a stressed semiconductor sample. E.A.C. and P.M-H. analyzed the data. E.A.C. drafted the manuscript with input from all authors. E.A.C. coordinated the project.

## Competing interests

The authors declare no competing interests.
