## [Peer Review File · Nature Communications]

REVIEWERS' COMMENTS

Reviewer #1 (Remarks to the Author):

I strongly recommend this paper for publication in Nature Communications. My arguments are formulated in the first report. In revised version authors fully match my recommendations on some technical improvement.

I read with great interest authors responds on the important questions raised by the second referee. I am convinced with these explanations. They clearly explains the importance of the achieved high polarization degree of the nuclear spin system. And also show the important problems to be solved in future.

Reviewer #2 (Remarks to the Author):

The authors have addressed all of my comments satisfactorily. I recommend publication for Nature Communications with a minor revision:

- It would be helpful if the authors could provide a reference for the calculations done for Eq. S15

Reviewer #3 (Remarks to the Author):

Reviewer #4 (Remarks to the Author):

RESPONSE TO REVIEWERS' COMMENTS

Reviewer #1 (Remarks to the Author):

I strongly recommend this paper for publication in Nature Communications. My arguments are formulated in the first report. In revised version authors fully match my recommendations on some technical improvement.

I read with great interest authors responds on the important questions raised by the second referee. I am convinced with these explanations. They clearly explains the importance of the achieved high polarization degree of the nuclear spin system. And also show the important problems to be solved in future.

We thank Reviewer #1 for their positive assessment.

Reviewer #2 (Remarks to the Author):

The authors have addressed all of my comments satisfactorily. I recommend publication for Nature Communications with a minor revision:

- It would be helpful if the authors could provide a reference for the calculations done for Eq. S15

We thank Reviewer #2 for their positive assessment. One of the authors (E.A.C) derived Eq. S15 analytically, using computer software (Wolfram Mathematica). It is possible that the same derivation had been carried out previously, but unfortunately, we are not aware of any previous publication and therefore cannot provide a reference. However, this derivation is not a major claim of our work. In the revised manuscript we provide a step-by-step derivation. Due to the size, this is presented in a separate file. Both the Wolfram Mathematica format file (DerivationVarianceM.nb) and a pdf version (DerivationVarianceM.pdf) are provided as Supplementary to the manuscript. Furthermore, the equation S15 is now written in a more compact form, using the hyperbolic cosecant functions.

Reviewer #3 (Remarks to the Author):

Reviewer #4 (Remarks to the Author):

We thank Reviewers #3 and #4 for their efforts.